# Bacterial Colonization of the Female Upper Genital Tract

**DOI:** 10.3390/ijms20143405

**Published:** 2019-07-11

**Authors:** Adriana Peric, Jürgen Weiss, Nicolas Vulliemoz, David Baud, Milos Stojanov

**Affiliations:** 1Clinic for Reproductive Medicine and Gynecological Endocrinology, Cantonal Hospital Lucerne, 6000 Lucerne, Switzerland; 2Fertility Medicine and Gynaecologic Endocrinology Unit, Department Woman-Mother-Child, Lausanne University Hospital, 1011 Lausanne, Switzerland; 3Materno-fetal and Obstetrics Research Unit, Department Woman-Mother-Child, Lausanne University Hospital, 1011 Lausanne, Switzerland

**Keywords:** upper genital tract, microbiota, 16S rRNA metagenomics, *Lactobacillus*

## Abstract

Bacteria colonize most of the human body, and the female genital tract is not an exception. While the existence of a vaginal microbiota has been well established, the upper genital tract has been considered a sterile environment, with a general assumption that bacterial presence is associated with adverse clinical manifestation. However, recent metagenomic studies identified specific patterns of microbiota colonizing the uterus, fallopian tubes, ovaries, and placenta. These results need confirmation and further investigations since the data are only scarce. Bacterial colonization of these sites appears different from the vaginal one, despite evidence that vaginal bacteria could ascend to the upper genital tract through the cervix. Are these bacteria only commensal or do they play a role in the physiology of the female upper genital tract? Which are the genera that may have a negative and a positive impact on the female reproductive function? The aim of this review is to critically present all available data on upper genital tract microbiota and discuss its role in human reproduction, ranging from the technical aspects of these types of analyses to the description of specific bacterial genera. Although still very limited, research focusing on genital colonization of bacteria other than the vaginal milieu might bring novel insights into physiopathology of human reproduction.

## 1. The Female Upper Genital Tract—Not as Sterile as Presumed?

With recent advancements in sequencing technologies, we have begun to appreciate the role of human microbiota and particularly bacteria, which are not passive commensals, but have a profound influence on host homeostasis. Indeed, bacteria play a significant role at multiple levels, including protection against pathogens, maturation of the immune system, metabolic pathways, vitamin synthesis, and others [1,2,3]. It is, thus, not surprising that an imbalance or dysbiosis of the microbiota has been associated with several adverse outcomes and diseases [2,4]. The genital system is certainly not an exception to bacterial colonization [5,6,7]. Aside from pathogenic bacteria that may be associated with adverse pregnancy outcomes or infertility, the role of physiological microbiota has been underestimated. Thanks to recent advancements, there is growing evidence of a relationship between microbiota and fertility, hormonal status, antibiotic treatment, as well as sexual habits [6,8,9,10,11,12,13].

The best-characterized genital niche from a microbial colonization point of view is the human vagina, which may be colonized by a wide variety of anaerobic and aerobic bacterial species, but which is physiologically dominated by presence of the *Lactobacillus* genus [7]. The activity of lactobacilli helps to maintain vaginal homeostasis through different mechanisms, including a reduction of the vaginal pH, production of active compounds against deleterious bacteria, physical barriers to adhesion, and stimulation of the host immune system against pathogens [14,15,16]. Disruption of vaginal homeostasis (i.e., reduction of *Lactobacillus* spp.) is characterized by increased bacterial colonization and establishment of a more diverse microbiota that could lead to adverse gynaecological and obstetrical outcomes, such as bacterial vaginosis and preterm labor [17,18,19]. 

Until recently, it was widely believed that the uterus was a sterile organ protected by the cervical mucus from ascending bacteria present in the vaginal milieu, especially during pregnancy. The mucosal epithelial cells of the cervix form a barrier by producing a highly hydrated gel that covers the cervical surface and protects them against chemical and mechanical injuries as well as microbial attacks [8,20]. These cells also secrete many defensive substances like mucins, defensins, protegrins, lysozyme, histatins, and nitric oxide [21]. The “sterile womb” hypothesis was challenged by studies showing that bacteria were nevertheless able to ascend through the cervical mucus plug [20]. Not only bacteria, but also small labelled particles, were observed to be actively transported from the posterior vaginal fornix, through cervix and uterus, and up to the fallopian tubes and the dominant follicle during follicular and luteal phases [22]. This is caused by spontaneous uterine contractions, with the highest frequency around time of ovulation and during orgasms that participate in the transport of spermatozoa to the fallopian tubes [23,24,25]. Bacterial colonization of the upper genital tract (UGT) is supported by several reports recently published [26,27,28] and differs from the vaginal bacterial content (Figure 1). While the impact of bacterial colonization on the human vagina has been extensively studied, there is only very limited understanding how different bacterial species could influence homeostasis of the UGT and have a positive or negative influence on reproduction. It is, therefore, of the uttermost importance to better characterize the role of the UGT microbiota, which is in close contact with the embryo during its development. In that respect, the endometrium milieu might play a significant role in regard to the significant difference of the anatomy, physiology, and immune response compared to the vagina.

## 2. Upper Genital Tract Microbiota Sampling and Challenges

Unlike vaginal samples, specimen collection from the UGT is more challenging and has several disadvantages (Table 1). Access to the vagina requires either self-sampling or a speculum exam and is, thus, simple to perform. On the other hand, UGT sampling is certainly more invasive, and samples are mainly obtained during surgical and explorative procedures. 

Initial analyses have highlighted that the microbiota abundancy of the UGT is significantly lower, compared to the vaginal one, by several orders of magnitude [29]. Consequently, possible contamination from the richly colonized lower tract is among the greatest concerns when collecting samples because even slight contaminations may render the analyses uninterpretable. Unless taking samples directly from the endometrium after hysterectomy [30] or during laparoscopic surgery [26], the vaginal–cervical route is the only option to reach the UGT. To avoid contamination, different sampling techniques and tools have been developed. Moreno et al. used the Novak curette, which is inserted through a vaginal speculum, under visual control to avoid any contact with the vaginal wall [31]. Since the Novak curette is connected to a 20 mL syringe, endometrial samples can be taken through suction. This has allowed the collection of a larger amount of material compared to other studies, which used an embryo transfer catheter tip to collect samples of the endometrium [28,32]. Another method to obtain endometrial samples is to aspirate endometrial fluid through a transcervically placed flexible transfer catheter connected to a syringe. To prevent contamination by cervical mucus during catheter removal, suction is stopped at the entrance of the internal cervical orifice and cervical mucus is aspirated prior to endometrial fluid aspiration. Approximately 20–40 μL of endometrial fluid have been previously obtained using this technique [27]. 

The Tao Brush IUMC Endometrial Sampler (Cook OB-GYN, Bloomington, IN, USA), which should decrease the risk of cervico-vaginal contamination, has been used by Verstraelen and colleagues. The brush is covered with a plastic protective sheath laterally and by a small plastic bead on top to completely protect the brush from contamination during passage through the vagina and endocervical canal. The brush is only unsheathed after passing the cervical canal and reaching the uterine cavity, and it is resheathed before withdrawal [33]. 

Another problem to face when studying ecological niches with low microbiota levels is the presence of bacterial DNA in the kits and materials that are used to process the samples. DNA extraction kits carrying bacterial DNA from water and soil origins have been documented [34]. Therefore, negative controls processed and sequenced in parallel with samples that allow the monitoring of basal contaminants should always be performed [35,36].

Interestingly, one study suggested that cervico-vaginal contamination should not play a crucial role in the validity of the results. Chen et al. compared endometrial samples taken during surgery from minimally invasive cuts in the uterus during laparoscopy or laparotomy and compared them to samples taken via the cervical orifice from the same patients [26]. The results were very similar, without any evidence of contamination from vaginal or cervical microbiota [26]. Therefore, sampling through the cervix should be acceptable for the analysis of the UGT microbiota when all the precautions to avoid contamination are taken, although additional studies should confirm this assumption.

Patient selection also has an important impact on the results of endometrial microbiota characterization. Sampling from the cervico-vaginal route is mainly performed for women in the reproductive phase, while endometrium sampling after hysterectomy is mainly performed in perimenopausal or postmenopausal women. As a consequence, the hormonal status of patients may influence the outcome of bacterial colonization, as shown for the vaginal microbiota [37]. As patients between studies or even within the same study differ regarding age, menopausal status, body mass index (BMI), parity, and gynaecological health [29,33,38], it might be difficult to compare the results of microbiota analyses. In younger women, hysterectomy is almost exclusively performed in cases of uterine pathologies like dysfunctional uterine bleeding, benign uterine tumours, or oncological conditions, which potentially also alter the intrauterine environment [33]. Since examination of the endometrial milieu is a much more invasive procedure compared to vaginal swab sampling, having a large sample size or including healthy patient controls remains a challenge.

## 3. Endometrial Microbiota

The endometrium plays, obviously, a key role in reproduction. It comprises complex physiological changes during the menstrual cycle of premenopausal women under the influence of estrogen and transformation into a receptive state under the influence of progesterone. It is also swayed by the complex interplay of growth factors, integrins, prostaglandins, and immune-modulating cytokines as well as angiogenesis [39]. Successful embryo implantation is dependent on the synchronized dialogue between a receptive endometrium and a normal developing embryo at the blastocyst developmental stage [40]. The causes of implantation failure still remain unclear in many cases. Since better understanding of implantation failure and its treatment could lead to higher implantation rates in *in vitro* fertilization (IVF) cycles, this topic is the focus of intensive research. There is rising evidence that endometrial microbiota could have an impact on the embryo-maternal interactions at the time of implantation; thus, most UGT microbiota report focus on the endometrium.

Table 2 summarizes all the studies performed on bacterial colonization of the endometrium. Interestingly, in the majority of studies, *Lactobacillus* was the most prominent genus. Several other genera were identified as colonizers of the endometrium, however, not as consistently as *Lactobacillus*. It has, therefore, been proposed to separate endometrial microbiota into *Lactobacillus*-dominated ones (more than 90% of *Lactobacillus* spp.) and non-*Lactobacillus*-dominated ones (less than 90% of *Lactobacillus* spp. and more than 10% of other genera) [27]. It is still unclear whether lactobacilli identified in the uterine cavity ascended from the vagina or if they are true colonizers of the endometrium. Despite multiple analyses by next generation sequencing (NGS), there is still not a consensus on the composition of the endometrial microbiota. Importantly, in multiple studies, lactobacilli were not the predominant bacteria [33,38,41]. Nevertheless, several points may indicate that *Lactobacillus* species are actually an important part of the endometrial microbiota: (1) lactobacilli were identified from the endometrium of women undergoing total hysterectomy, thus eliminating the vaginal contamination problem [30]; (2) lactobacilli were both identified from the endometrium sample and endometrial fluid [27,42]; and (3) lactobacilli were the major genus using different sequencing techniques and diverse variable regions of the 16S rRNA gene [27,30,32]. The activity of lactobacilli helps to maintain the natural, healthy balance of the vaginal microbiota through different mechanisms, including a reduction of the vaginal pH, production of active compounds against deleterious bacteria, physical barriers to adhesion, and stimulation of the host immune system against pathogens [14,16]. Yet, the uterine cavity displays different characteristics from the vagina, and it is not clear how these bacteria could promote endometrial homeostasis and positively influence embryo implantation.

In a pioneering study, Mitchell et al. analyzed the bacterial content of the uterus from 58 patients undergoing hysterectomy using real-time quantitative PCR to detect 12 bacterial species known to colonize the vagina [29]. Bacteria were detected in most women (95%), and the most prevalent ones were *Lactobacillus iners* (45% of samples), *Prevotella* spp. (33% of samples), and *L. crispatus* (33% of samples). They also observed that the bacterial content was significantly lower compared to the vagina. 

With the advancements in sequencing techniques, the uterine microbiota has been analyzed for several reasons in order to assess any possible role of endometrial bacteria on specific outcomes or circumstances. Although with some exceptions, *Lactobacillus* was widely identified as the most abundant colonizer of the endometrium. By examining bacterial content of the most distal portion of the embryo transfer catheter tip, Franasiak et al. did not observe any significant difference between the microbiota of subsequent pregnant (n = 18) and nonpregnant (n = 15) women, highlighting *Lactobacillus* as the most abundant genus [28]. In addition, *Flavobacterium* spp. were also found to dominate endometrial microbiota. Similarly, *Lactobacillus* spp. were predominant in a study comprising 70 women undergoing embryo transfer, in which catheter tips were again used for the analysis. This genus’ abundance was over 70% in 50 samples. Other genera consistently identified in this study were *Corynebacterium* (40 patients), *Bifidobacterium* (15 patients), *Staphylococcus* (38 patients), and *Streptococcus* (38 patients) [32]. Similar studies on women seeking IVF have focused on endometrial fluid instead of endometrial tissue. Moreno et al. compared vaginal and endometrial fluid microbiota and concluded that, generally, *Lactobacillus* dominated in both organs, although differences between the two sites were observed in some subjects [27]. The authors divided the samples into two groups, *Lactobacillus*-dominated microbiota (>90% *Lactobacillus* spp.) and non-*Lactobacillus*-dominated microbiota (<90% *Lactobacillus* spp. with >10% of other bacteria), based on a definition from a previous analysis of vaginal microbiota [46]. Interestingly, patients with a non-*Lactobacillus*-dominated microbiota showed significantly increased adverse reproductive outcomes when compared to patients in which *Lactobacillus* dominated the endometrial microbiota [27]. They also observed that bacterial composition of endometrial fluid did not change between the pre-receptive and receptive phases of the endometrium, suggesting that endometrial microbiota remained stable and were not hormonally dependent. Nevertheless, analysis of the endometrial tissue should be performed to fully support this conclusion. By analyzing endometrial fluid, Kyono and colleagues observed that incidence of non-*Lactobacillus*-dominated microbiota was significantly higher in patients undergoing IVF compared to healthy volunteers and non-IVF patients, suggesting again that lactobacilli may be beneficial in the initial phases of embryo development [45]. Nevertheless, endometrial microbiota of IVF patients may be negatively influenced by the procedures performed during infertility investigation and treatment. In a following study, the same authors sought to interfere with non-*Lactobacillus*-dominated endometrial microbiota using antibiotics and probiotics [42]. However, even in the case of the restoration of *Lactobacillus*-dominated microbiota, pregnancy rates after single vitrified-warmed blastocyst transfer failed to show any significant results [42]. 

Despite the fact that most studies highlight high levels of *Lactobacillus* colonization in the endometrium, a study analyzing samples from 19 women with reproductive failures (recurrent implantation failures, recurrent pregnancy losses, or both) has shown that the majority of colonizing bacteria belonged to the Bacteroidetes and Proteobacteria phyla. More specifically, these bacteria belonged to *Bacteroides* (*B. xylanisolvens*, *B. thetaiotaomicron*, and *B. fragilis*) and *Pelomonas* genera, respectively. Presence of *Lactobacillus* spp. as the dominant genus was observed in four patients, of which two were *L. crispatus* and the other two *L. iners*. Uterine anomalies were excluded by hysteroscopy in these patients, which included women with recurrent implantation failure and recurrent pregnancy loss [33]. The absence of healthy controls does not rule out the assumption that endometrial colonization of these patients could be biased by the procedures performed during the IVF processes. 

The role of the endometrial microbiota has been investigated not only in infertile patients but also in different gynaecological diseases. A pilot study performed by Walther-António et al., compared the microbiota of 17 patients with endometrial cancer with that of patients with endometrial hyperplasia (a precancerous state) and a control group with benign uterine conditions [38]. The study involved sampling of vagina, cervix, uterus, fallopian tubes, ovaries, and stool after hysterectomy. The results demonstrated that not only were microbiota of different organs significantly correlated with respect to the sampling site, but that specific bacteria were significantly enriched in the endometrial cancer cohort. *Escherichia*, *Streptococcus*, *Staphylococcus*, and *Enterococcus* were the predominant genera in patients with chronic endometritis and dysfunctional bleeding. The simultaneous endometrial presence of *Atopobium vaginae* and a previously uncultured representative of the *Porphyromonas* genus was associated with endometrial cancer, particularly if connected to high vaginal pH > 4.5. Additional studies on larger cohorts should confirm these putative markers of endometrial cancer and their clinical relevance using healthy controls as comparison (patients with benign uterine conditions such as abnormal bleeding, fibroids, prolapse, and pelvic pain). While lactobacilli could be identified in multiple patients, they were far from being the predominant bacteria in all types of samples, thus not agreeing with the majority of studies exploring endometrial microbiota. Involvement of uterine microbiota has been investigated for endometrial polyps, which are a possible feature of chronic endometritis [41]. The intrauterine microbiota differed significantly from the vaginal microbiota from the same patient. Surprisingly, increased incidence of *Lactobacillus*, as well as *Bifidobacterium*, *Gardnerella*, *Streptococcus*, and *Alteromonas*, was associated with the appearance of endometrial polyps. Given the low number of symptomatic patients enrolled in the study (n = 20), more evidence is needed to support these conclusions.

Endometrial microbiota has been analyzed in woman with endometriosis [43]. Among both controls (n = 32) and patients with endometriosis (n = 32), five predominant bacterial families were observed (i.e., *Lactobacillacae, Streptococcaceae, Staphylococaceae, Enterobacteriaceae, and Moraxellaceae*). Patients with endometriosis showed a significant increase of *Streptococcaceae* and *Moraxellaceae* colonization, while a decrease, although statistically nonsignificant, of *Lactobacillacae* was observed when compared to controls. Patients treated with gonadotropin releasing hormone agonist (GnRHa), used as treatment for endometriosis, displayed a significantly decreased *Lactobacillacae* and increased *Streptococcaceae*, *Staphylococaceae*, and *Enterobacteriaceae* colonization, indicating that hormonal changes could influence endometrial microbiota [43].

Pelzer and colleagues performed an analysis of bacterial colonization of the endometrium in patients with menorrhagia or dysmenorrhea [44]. The study included samples during different menstrual cycle phases of 145 women undergoing operative hysteroscopy or laparoscopy. The authors concluded that the endometrial microbiota was dominated by the *Lactobacillus* genus, and that abundance of bacteria belonging to *Propionibacterium*, *Staphylococcus*, *Pseudomonas*, *Ralstonia*, *Corynebacterium*, and *Kocuria* genera was significantly increased compared to the endocervix. Moreover, microbiota significantly changed between the secretory (increased abundance of *Prevotella* spp.) and proliferative phases of the menstrual cycle (increased abundances of *Sneathia* spp. and *Lactobacillus* spp.).

## 4. Bacterial Colonization of Other Sites of the Upper Genital Tract (UGT)

There is now evidence that there is a microbiota continuum along the entire female genital tract (Table 3). In an initial attempt to characterize microbiota of multiple sites, Miles and colleagues performed NGS on samples coming from women undergoing a total hysterectomy and bilateral salpingo-oopherectomy [30]. This pilot study included 10 patients and focused on the vagina, cervix, endometrium, myometrium, fallopian tubes, and ovaries. Despite several limitations, like the small sample size, varying surgical procedure routes, and women in both reproductive and menopausal stages, this study has shown that bacteria of the myometrium, endometrium, and fallopian tubes differed significantly compared to the vaginal content, and that the UGT microbiota was generally more diverse compared to the vagina. A seminal study including 110 patients with the same goal was performed shortly after [26]. The authors confirmed that microbial communities diverge from the vagina, cervical canal, uterus, fallopian tubes, and peritoneal fluid. The number of uterine bacteria was estimated to be about four orders of magnitude lower than vaginal bacteria. Vaginal microbiota had a typical signature composed of *L. iners* and *L. crispatus*, but interestingly a core microbiota could not be identified for the endometrium. Instead, specific recurrent taxa were found in the fallopian tubes, including *Pseudomonas*, *Erysipelothrix*, and *Facklamia*. In addition, the peritoneal fluid (obtained through the Douglas pouch) featured a core microbiota composed of *Pseudomonas*, *Morganella*, *Sphingobium*, and *Vagococcus*. Therefore, a clear distinction between the lower and the upper genital tract could be observed, with the cervix and endometrium as the transition zone. Moreover, observations made during this study indicated that composition of the vaginal and uterine microbiota could predict adverse outcomes (hysteromyoma, adenomyosis, and endometriosis) in women of reproductive age and, thus, could be used as putative markers, but larger cohorts are needed to confirm this hypothesis.

Fallopian tube microbiota was assessed in 16 patients, half of which were in the pre- and the other half in postmenopausal stage, by Pelzer et al. [47]. The predominant taxa were *Staphylococcus* spp., *Enterococcus* spp., and *Lactobacillus* spp. Other recurrent genera included *Pseudomonas* spp., *Burkholderia* spp., *Propionibacterium* spp., and *Prevotella* spp. Interestingly, bacterial colonization was different between right and left tubes, with *Staphylococcus* spp. more abundant on the right, while *Lactobacillus* spp., *Enterococcus* spp., and *Prevotella* spp. were more abundant on the left. Moreover, bacterial communities were different in pre- and postmenopausal women, thus indicating a possible role of hormonal regulation on the microbiota. 

Bacterial communities of the ovary were explored in patients with ovarian cancer, with the aim to determine bacterial markers of tumours. In one study, *Acinetobacter*, *Sphingomonas*, and *Methylobacterium* genera were significantly enriched in samples from patients with cancer, whereas *Lactococcus spp.* was highly enriched in the control group [48]. In a second study, a large number of bacterial genera was associated with the disease. The authors support the hypothesis that the tumour microenvironment creates a niche in which multiple types of bacteria can bloom [49].

Existence of placental microbiota has been debated since its initial description in 2014. Aagaard et al. described the placental microbiota as similar to the oral microbiota, composed by several genera of the Firmicutes, Tenericutes, Proteobacteria, Bacteroidetes, and Fusobacteria phyla [50]. While different studies have confirmed this observation [52,54,55], others indicated that bacterial content of the placenta was not significantly different from the negative controls included in the analyses [56,57,58,59]. Such divergences originating from the very low bacterial load in this organ and the choice of the variable portion of the 16S rRNA gene that are included in the analyses (V1–V2 region primers versus V4 region primers in this case) may give different outcomes, as suggested by Seferovic et al. [53]. Importantly, this study confirmed that bacteria actually colonized the placenta, by combining in situ hybridization of 16S rRNA probes to detect single bacterial cells in parallel to metagenomic analysis [53]. Several other studies have found the association in placental microbiota shifts and adverse outcomes, including preterm birth, gestational diabetes, and excess weight increase [51,55,60,61], highlighting a potentially important role of these bacteria.

## 5. Experimental Assessment of the Role of Uterine Microbiota: Animal and in Vitro Models

Recent metagenomic studies allowed establishment of the concept that the uterine cavity and other sites of the UGT harbour their own microbiota. Therefore, future studies should not only focus on the description of the colonizers but furthermore assess bacterial role in UGT homeostasis. This is, however, not a trivial task. Initially, specific bacteria identified by metagenomic analyses must be isolated and cultivated from clinical samples. This, besides the improvement of culturing techniques, remains an important limitation. Moreover, in opposition to studies implying a “one-pathogen model”, colonizers may form complex communities, in which specific bacteria may require the presence of others for their survival.

Animal models have been largely used to study microbiota, especially in the case of human gut microbiota [62]. Concerning bacteria colonizing the genital tract, murine vaginal and uterine models of infection have been used to study human pathogens like *Chlamydia trachomatis* [63,64,65], *Mycoplasma genitalium* [66,67,68], or *Neisseria gonorrhoeae* [69,70,71]. However, while useful to dissect the infectious process, murine genital models have important physiological differences compared to the human vagina, which in its healthy state is colonized mainly by *Lactobacillus spp.* and characterized by a low pH (median pH of 4.5 compared by 5.4–7.8 among other mammals). What is the impact of these important differences in the pathogenesis of human isolates in animals? Interestingly, *C. trachomatis* is quickly cleared if inoculated in mouse vagina, while its murine counterpart, *Chlamydia muridarum*, is capable of inducing similar gynecological outcomes in mice as those observed in infected patients [63,72]. In order to use animal models to study UGT microbiota, additional studies of their uterine microbiota are required. So far, there are no reports on microbiota of the murine UGT. Interestingly, embryo transfer in germ-free mice resulted in a lower rate of success compared to wild-type mice. This type of experimental setting could be used to evaluate the positive or negative outcomes of specific bacteria or more complex bacterial communities.

On the other side, uterine microbiota has been well-characterized in cows and, more recently, in dogs [73,74,75,76]. Bovine infertility due to endometritis results in great economic losses and is the reason why endometrial microbiota has been investigated. Similar to human studies, an endometrial core microbiota could not be defined, as different results were observed within different reports [73,74,75]. In the unique report about dog uterine colonization, *Pseudomonas*, *Staphylococcus*, and *Campylobacter* were the predominant genera. Interestingly, bacteria colonizing the human endometrium were retrieved, although at a very low abundance [76]. 

*In vitro* models might represent an interesting alternative to animal experimentation to initially explore the uterine microbiota and interaction between endometrium and bacteria of this niche. While a simple cellular layer consisting of endometrium-derived cells might be too simplistic to analyze complex interactions, novel models more relevant from morphological and physiological points of view are becoming available. Among them, endometrial air–liquid interface cultures and endometrial organoids have great potential in the experimental assessment of the impact of bacterial colonization of the UGT [77,78,79,80]. While physiologically relevant, these closed systems allow tight regulation of many parameters, including pH, CO_2_ partial pressure, and nutriments, among others. Ideally, combinations of *in vitro* and in vivo approaches may help in understanding the role of specific bacteria colonizing UGT. Figure 2 depicts the main strength and disadvantages of the model presented above.

## 6. Conclusions

The field of UGT microbiota is still in the early days of development, and its physiological role in human reproduction remains to be understood. Unlike the vagina, UGT sample collection is complex, invasive, and available data showed a low abundant and highly diverse microbiota. None of the studies conducted so far allowed to characterize a well-defined core microbiota of the uterus, but altogether they challenged the “sterile womb” dogma. It is now clear that bacteria are in close contact with the embryo, even during the early stages of development, and additional efforts should be done to assess their role. Are these microorganisms only passive commensals or do they have a role beyond infection by interfering positively or negatively with human reproduction? What is the impact of hormonal regulation on UGT microbiota? Are there any bacterial markers that could predict fertility potential, especially in infertile patients seeking assisted reproduction? Is there any relevant bacterial signature allowing early detection of pathologies? The challenge for future metagenomic studies is utilization of larger cohorts and standardization of methods. The high variability of colonizing bacteria could be the result of different sampling strategies and different characteristics of patients enrolled in the analysis (age, reproductive stage, or pathologies). Importantly, not only do metagenomic analyses of different variable regions of the 16S rRNA show discrepancies in results, but utilization of different sequencing techniques could be a great bias for comparison of different studies. Is the endometrial microbiota highly diverse, or are the disagreements between studies artefacts caused during analyses?

With few exceptions, *Lactobacillus* is the sole genus consistently found in the endometrium, and some studies have found a positive association with the reproductive outcome. The question of whether the presence of uterine lactobacilli reflects only a healthy vaginal milieu simply crossing the cervical barrier or has a physiological role in the uterus should be assessed by rigorous experiments.

It will be crucial to select relevant animal models coupled to innovative *in vitro* approaches to study the impact of UGT bacteria on reproduction. An effort towards the characterization of the reproductive microbiota of the most used animal models, including rodents, rabbits, or swine, should be performed in order to select the model that is the closest to the human endometrium. 

The fascinating impact of bacterial colonisation of the human body has only been untangled during the last decade. Functional characterization of complex bacterial communities is certainly not a trivial task, but, given the growing number of infertile patients seeking treatment, an effort towards understanding the role of genital microbiota is paramount, especially in the initial phases of embryo development within the UGT.

## Figures and Tables

**Figure 1 ijms-20-03405-f001:**
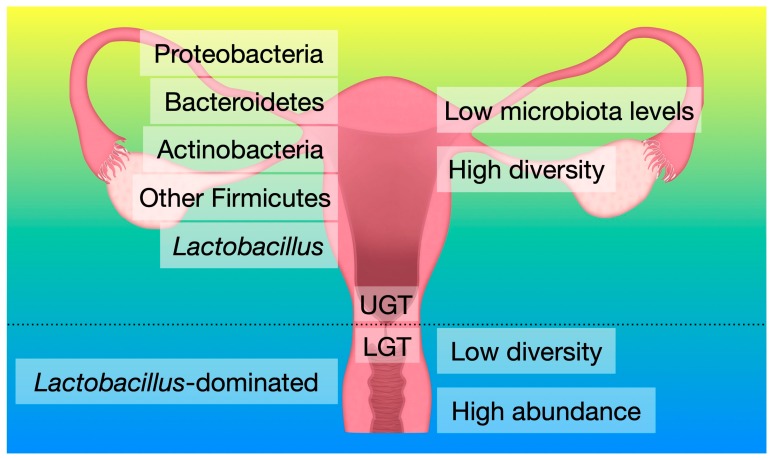
General characteristics of bacterial communities of the lower genital tract (LGT) and the upper genital tract (UGT).

**Figure 2 ijms-20-03405-f002:**
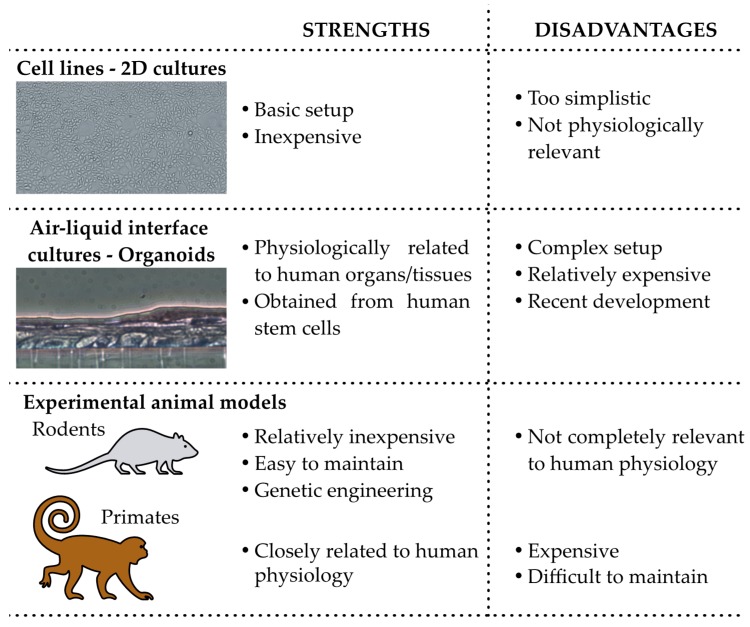
Strengths and disadvantages of the experimental models that can be used to assess the impact of the microbiota.

**Table 1 ijms-20-03405-t001:** Main experimental differences between the analysis of vaginal (lower genital tract) and uterine (upper genital tract) microbiota.

Characteristic	Vagina (Lower Genital Tract)	Uterus (Upper Genital Tract)
Sampling procedure	• Low invasiveness • Simple (vaginal swab)	• Invasive methods • Through the vaginal–cervical route, after hysterectomy or during laparoscopic surgery
Target patients	• Can be routinely performed in all type of patients	• Patients with endometrial pathologies (endometriosis or cancer) or reproductive issues
Contamination risk during sampling	• Low risk of contamination	• Risks of contamination by vaginal microbiota when accessing uterus through the vaginal–cervical route
Data interpretation	• Well-defined vaginal microbiota based on a large number of studies	• Scarcely studied • Comparison between studies is difficult due to the heterogeneity of targeted patient populations • Difficult to obtain healthy controls

**Table 2 ijms-20-03405-t002:** Metagenomic studies performed on endometrial samples.

Study	Study Targets	Sample Type	Sequencing Method and Target	Country	Major Genera	Impact on the Outcome ^1^	Ref.
Franasiak et al. (2016)	Patients undergoing *in vitro* fertilization (IVF) (n = 33) • 18 in ongoing pregnancies • 15 without pregnancies	Embryo transfer catheter tips	Ion Torrent • V2,3,4,6,7,8,9	USA	*Flavobacterium* spp.	NA	[28]
					*Lactobacillus* spp.	NA	
					*Limnohabitans* spp.	NA	
					*Polynucleobacter* spp.	NA	
					*Bdellovibrio* spp.	NA	
					*Chryseobacterium* spp.	NA	
					*Spirochaeta* spp.	NA	
					*Clostridium* spp.	NA	
Verstraelen et al. (2016)	Patients undergoing hysteroscopy (n = 19) • 11 with recurrent implantation failure • 7 recurrent pregnancy loss • 1 with both	Endometrial sampler	Illumina • V1–V2	Belgium	*Bacteroides* spp. (dominant)	NA	[33]
					*Lactobacillus crispatus*	NA	
					*Pelomonas* spp.	NA	
					*Prevotella* spp.	NA	
					*Atopobium vaginae*	NA	
					*Mobiluncus curtisii*	NA	
Khan et al. (2016)	Patients with endometriosis (n = 32) • 16 with endometriosis • 16 controls without endometriosis	Endometrium and cystic fluid	Illumina • Custom primers	Japan	*Lactobacillacae*	+	[43]
					*Streptococcaceae*	−	
					*Staphylococaceae*	−	
					*Enterobacteriaceae*	NA	
					*Moraxellaceae*	−	
Fang et al. (2016)	Patients with endometrial polyps (n = 30) • 20 with endometrial polyps • 10 healthy women	Endometrial swabs and endometrial tissues	Illumina • V4	China	*Lactobacillus*	−	[41]
					*Bifidobacterium*	−	
					*Gardnerella*	−	
					*Streptococcus*	−	
					*Alteromonas*	−	
					*Enterobacter*	+	
					*Pseudomonas*	+	
Moreno et al. (2016) *	Patients undergoing IVF (n = 70) • 35 undergoing IVF • 35 fertile women	Endometrial fluid	454 pyrosequencing • V3–V5	Spain	*Lactobacillus* spp. (dominant)	+	[27]
					*Gardenrella* spp.	−	
					*Bifidobacterium* spp.	−	
					*Streptococcus* spp.	−	
					*Prevotella* spp.	−	
Walther-António et al. (2016) *	Women undergoing hysterectomy (n = 31) • 17 with endometrial cancer • 4 with endometrial hyperplasia • 10 with benign gynecologic condition	Uterine biopsies	Illumina • V3–V5	USA	*Shigella* spp.	NA	[38]
					*Barnesiella* spp.	NA	
					*Staphylococcus* spp.	+	
					*Blautia* spp.	+	
					*Parabacteroides* spp.	+	
					*Bacteroides* spp.	−	
					*Faecalibacterium* spp.	−	
Miles et al. (2016) *	Women undergoing a total hysterectomy and bilateral salpingo-oopherectomy (n = 10) • Patients with various pathologies	Endometrium tissue	454 pyrosequencing • V1–V3	USA	*Lactobacillus* spp.	NA	[30]
					*Corynebacterium* spp.	NA	
					*Staphylococcus* spp.	NA	
					*Acinetobacter* spp.	NA	
					*Blautia* spp.	NA	
Tao et al. (2017)	Patients undergoing IVF (n = 70)	Embryo transfer catheter tips	Illumina • V4	USA	*Lactobacillus* spp. (dominant)	NA	[32]
					*Corynebacterium* spp.	NA	
					*Bifidobacterium* spp.	NA	
					*Staphylococcus* spp.	NA	
					*Streptococcus* spp.	NA	
Chen et al. (2017) *	Patients without infection (n = 110)	Endometrium	Ion Torrent • V4–V5	China	*Lactobacillus* spp. (30.6%)	NA	[26]
					*Pseudomonas* spp. (9.09%)	NA	
					*Acinetobacter* spp. (9.07%)	NA	
					*Vagococcus* spp. (7.29%)	NA	
					*Sphingobium* spp. (5%)	NA	
Pelzer et al. (2018)	Patients undergoing operative hysteroscopy or laparoscopy (n = 145)	Endocervical swabs	454 pyrosequencing • V5 and V8	Australia	*Lactobacillus* spp. (dominant)	NA	[44]
					*Gardnerella* spp.	NA	
					*Veillonella* spp.	NA	
					*Prevotella* spp.	NA	
					*Sneathia* spp.	NA	
					*Jonquetella* spp.	NA	
					*Fusobacterium* spp.	NA	
Kyono et al. (2018) *	Patients with infertility (n = 109) • 79 undergoing IVF • 23 non-IVF cases • 7 fertile volunteers	Endometrial fluid	Illumina • V4 with modified primers	Japan	*Lactobacillus* spp.	+	[45]
					*Gardnerella* spp.	NA	
					*Streptococcus* spp.	NA	
					*Atopobium* spp.	NA	
					*Bifidobacterium* spp.	NA	
					*Sneathia* spp.	NA	
					*Prevotella* spp.	NA	
					*Staphylococcus* spp.	NA	
Kyono et al. (2019)	Patients undergoing IVF (n = 92) • 56 *Lactobacillus*-dominant microbiotas • 36 non-*Lactobacillus*-dominant microbiotas	Endometrial fluid	N/A	Japan	*Lactobacillus* spp.	+	[42]
					*Atopobium* spp.	NA	
					*Bifidobacterium* spp.	+	
					*Gardnerella* spp.	NA	
					*Megasphaera* spp.	NA	
					*Sneathia* spp.	NA	
					*Prevotella* spp.	NA	

* Vaginal samples have been analyzed in parallel; ^1^ Indicates if the taxon is positively (+) or negatively (−) associated with the study question (NA, not available).

**Table 3 ijms-20-03405-t003:** Metagenomic studies performed on non-endometrial samples of the female UGT.

Study	Study Targets	Sampling	Sequencing Method and Target	Country	Major Taxa	Ref.
**Fallopian Tubes**						
Miles et al. (2016)	Women undergoing a total hysterectomy and bilateral salpingo-oopherectomy (n = 10) • Patients with various pathologies	Ampullary region of the transected fallopian tubes	454 pyrosequencing • V1–V3	USA	Firmicutes	[30]
					Proteobacteria	
					Actinobacteria	
					Bacteroidetes	
Chen et al. (2017)	Patients without infection (n = 110)	Laparoscopy and laparotomy	Ion Torrent • V4–V5	China	*Acinetobacter* spp.	[26]
					*Comamonas* spp.	
					*Pseudomonas* spp.	
					*Dysgonomonas* spp.	
					*Vagococcus* spp.	
					*Delftia* spp.	
Pelzer et al. (2018)	Women undergoing total hysterectomy (n = 16)	Fallopian tube dissection	454 pyrosequencing • V5–V8	Australia	*Staphylococcus* spp.	[47]
					*Enterococcus* spp.	
					*Lactobacillus* spp.	
					*Pseudomonas* spp.	
					*Burkholderia* spp.	
					*Propionibacterium* spp.	
Zhou et al. (2019)	Women diagnosed with high-grade serous ovarian cancer (n = 50) • 25 cases • 25 controls	Distal fallopian tube tissues (used as control)	Illumina • V3–V4	China	Proteobacteria	[48]
					Firmicutes	
					Bacteroidetes	
					Actinobacteria	
**Ovary**						
Miles et al. (2016)	Women undergoing a total hysterectomy and bilateral salpingo-oopherectomy (n = 10) Patients with various pathologies	Ovary dissection	454 pyrosequencing • V1–V3	USA	Firmicutes	[30]
					Proteobacteria	
					Actinobacteria	
					Bacteroidetes	
					Verrucomicrobia	
Banerjee et al. (2017)	Women with ovarian cancer (primary and recurrent or metastatic tumors of ovarian origin, n = 139) • 99 ovarian cancer samples • 20 matched • 20 unmatched control	Ovarian cancer tissues	Illumina • Custom	USA	Proteobacteria	[49]
					Firmicutes	
					Bacteroidetes	
					Chlamydiae	
					Spirochaetes	
					Tenericutes	
Zhou et al. (2019)	Women diagnosed with high-grade serous ovarian cancer (n = 50) • 25 cases • 25 controls	Ovarian cancer tissues	Illumina • V3–V4	China	Proteobacteria	[48]
					Firmicutes	
					Bacteroidetes	
					Actinobacteria	
**Placenta**						
Aagaard et al. (2014)	Women at term and preterm (n = 48) • One-third with preterm delivery • One-third with remote antenatal infection • One-third women delivering at term	Placental sections located 4 cm from the cord insertion site	Illumina • V1–V3	USA	*Escherichia coli*	[50]
					*Prevotella tannerae*	
					*Bacteroides* spp.	
					*Streptomyces avermitilis*	
					*Propionibacterium acnes*	
					*Rhodococcus erythropolis*	
					*Neisseria polysaccharea*	
					*Neisseria lactamica*	
					*Fusobacterium* spp.	
Doyle et al. (2017)	Rural pregnant women (n = 1097)	Fetal (chorionic and amniotic) membrane and placental tissue near the umbilical cord insertion	Illumina • V5–V7	Malawi	*Acinetobacter* spp.	[51]
					*Enterobacteriaceae*	
					*Prevotella copri*	
					*Lactobacillus iners*	
Parnell et al. (2017)	Women at term (n = 57)	Samples from the placental villous, fetal membrane, and the basal plate	Illumina • V1–V9	USA	*Ralstonia insidiosa*	[52]
					*Mesorhizobium* spp.	
					*Lactobacillus crispatus*	
					*Lactobacillus iners*	
					*LUreaplasma nucleatum*	
Seferovic et al. (2019)	Women at term and preterm (n = 53) • 26 term deliveries • 26 preterm deliveries • 1 positive control	Dissected placental core sections	Illumina • V4	USA	Firmicutes	[53]
					Proteobacteria	
					Actinobacteria	
					Bacteroidetes	

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
