# Peer review of "Bacterial Colonization of the Female Upper Genital Tract"

_ijms, 2019, doi:10.3390/ijms20143405_

Reviewer 1 Report

Authors give a comprehensive review about bacterial colonization in the female UGT. The review is clearly written and will be very helpful to the field. 

Minor concern:

Line 220 of Page 5: Change vaginal pH>4,5 to vaginal pH>4.5

Author Response

We thank Reviewer 1 for the appreciation of our review. 

The mistake at line 224 was corrected as suggested.

Please find the revised manuscript for details.

Reviewer 2 Report

Dear Authors,

Thank you for your lots of works to summarize for the bacterial colonization in female upper genital tract.

The manuscript is well organized to understand the current studies on microorganisms in upper genital tract. However, the manuscript contained lots of studies so it is slightly hard to see the entire studies. So, you may propose a table to summarize the studies (for example table 1 you described in line 139) but I can't find the table. So, please let me see the table.

And, as a same reason, it will be great if you can give a summary table for session 4, "Not only in the uterus".

Title 4., "Not only in the uterus in line 249", seems not relevant to summarize the contents. So, please consider to change the title.

Minor comment.

Line pH 4,5 should be changed to pH 4.5.

Thank you.

Author Response

We thank Reviewer 2 for the positive comments on our manuscript.

Here are the details of the corrections:

1) We are sorry that table 1 was not displayed in the manuscript. We have added it after the reference section.

2) We agree that a table 2 on bacteria of the remaining parts of UGT will help the readership. Please find it in the manuscript (line 256).

3) We changed title of section 4 in "Bacterial colonization of other sites of the UGT". We hope that it describes better the content of the section.

4) The mistake at line 224 was corrected.

Round  2

Reviewer 2 Report

Dear Authors,

Thank you for the revisions and answered all the requests/questions from reviewers.

I'd like to thank you for the hard works for this manuscript specially on table 1.

Thank you.

Author Response

We thank the reviewer for the appreciation of our work.

Sincerely yours,

M. Stojanov